# Separating the generational effects of temperature and viscosity on the body size of a freshwater *Mesocyclops* copepod

Zachary Wagner[1,2,*], Griffin Wagner[1,3], David M. Fields[2] and Jeannette Yen[1]

## ABSTRACT

Copepods are small crustacean zooplankton that closely follow Bergmann's rule, which states that larger organisms will be found at higher latitudes. While thermally driven metabolic effects and food availability are often stated to be the major drivers behind this trend, temperature affects multiple variables within the copepod and its environment, a key variable to copepod ecology being viscosity. To test the effects of viscosity on copepod body size, two lineages of subtropical, freshwater *Mesocyclops* sp. copepods were grown for five generations in cultures of differing temperatures, 30°C and 18°C, and viscosities, natural and altered to mimic 18°C while at 30°C. Copepods grown at 30°C were, on average, 13.20% smaller than those grown at 18°C, regardless of viscosity. Copepods grown in 30°C cultures with a viscosity of 18°C had no body size differences when compared to copepods grown at 30°C and a natural viscosity. Copepods reached sexual maturity after 10 days while grown at 30°C and after 13 days while grown at 18°C, with viscosity playing no role in maturation time. As such, this study provides further support for temperature driving copepod body size, with the viscosity of the environment playing no discernible role in the body size of these small organisms.

KEY WORDS: Copepod, Zooplankton, Generation, Size, Temperature, Viscosity

## INTRODUCTION

One of the earliest theories in ecogeography is that of Bergmann's rule, which states that larger individuals of similar species will be found closer to the poles and smaller individuals will be found closer to the equator (Bergmann, 1847; James, 1970; Watt et al., 2010). The original focus on mammals has been expanded in recent years to include many other organisms where this trend in body size is notable, such as copepods, pelagic fish, ciliates, and phytoplankton (Atkinson et al., 2003; Saunders and Tarling, 2018; Evans et al., 2020; Campbell et al., 2021; Liu et al., 2023). Among these listed organisms, small organisms such as ciliates and copepods can provide valuable insight due to the complex relationship between their physiology and the physical properties of water.

Copepods are small, invertebrate holoplankton and the most numerous animals on the planet (Humes, 1994). They form the basis of almost every aquatic ecosystem by bridging primary producers to apex predators through other zooplankton (Brandl, 2005; Benedetti et al., 2016). The size of copepods, and of organisms generally, is crucial to their ecological function. Predators tend to target prey of an optimal size for their hunting and feeding strategies, which places focused pressure on the size structure of the community (Yen, 1985; Emmerson and Raffaelli, 2004). For instance, large predatory fish tend to have broader prey selection than smaller fish due to a larger mouth gape (Scharf et al., 2000). Similarly, predatory copepods are size-selective, with feeding rates generally showing an inverse relationship with prey size once the prey are larger than optimal (Yen, 1983, 1985; Gliwicz and Umana, 1994). As such, a decrease in the size of predatory copepods would lead to a downstream change in the prey they consume, and conversely, the former prey would face a different environmental pressure.

Temperature is the most likely driver of copepod body size, as copepod body length has a strongly correlated inverse relationship with temperature, with larger copepods being found in colder waters (Breteler and Gonzalez, 1988; Ban, 1994; Gaudy and Verriopoulos, 2004; Garzke et al., 2015; Sasaki et al., 2019; Roman and Pierson, 2022; de Juan et al., 2023). This relationship is consistent even within species that have multiple generations per year. For example, *Acartia tonsa* collected from the Berre Lagoon in Southern France in November were 33% larger than those collected in June (Gaudy and Verriopoulos, 2004). Copepods of the same species with a broad distribution range follow a similar trend, such as *A. tonsa* collected from Connecticut waters being larger than those from Florida, which correlates with colder water temperatures at higher latitudes (Sasaki et al., 2019). Temperature has a strong effect on copepod metabolism, which in turn dictates behavior and reproduction. High temperatures are often associated with increases in respiration rates, decreases in egg clutch size and individual egg fat content, and lowered egg hatching success (Ban, 1994; Doan et al., 2019; Heine et al., 2019). Lower temperatures are linked with longer embryonic development times but also larger female body sizes, which may allow for the storage of more fat for egg development (Ban, 1994; Ju et al., 2011).

While temperature has been shown to control different aspects of a copepod's biology, it additionally has an exponential, inverse relationship with the viscosity of a fluid. The lower the temperature is, the higher the viscosity, with the change in viscosity weakening as temperature increases. The Reynolds number is a measure of how much an organism is subject to viscosity; it is the ratio between the inertial forces (density of the fluid, flow speed, and length of the organism) and the viscous forces (dynamic viscosity of the liquid). This implies that a smaller and slower organism is more subject to the viscous forces of the liquid, while a larger and faster organism is

[1]Georgia Institute of Technology, School of Biological Sciences, Atlanta, Georgia, 30332, USA. [2]Bigelow Laboratory for Ocean Sciences, East Boothbay, Maine, 04544, USA. [3]University of Colorado Boulder, College of Arts and Sciences, Boulder, Colorado, 80309, USA.

*Author for correspondence (zwagner7@gatech.edu)

Z.W., 0009-0007-0200-3300; D.M.F., 0000-0002-8291-912X; J.Y., 0000-0002-0883-1306

more subject to turbulent forces. Copepods are unique among organisms in that they have an intermediate Reynolds number, being subject to both viscous and turbulent forces depending on their swimming modes and body size (Yen, 2000; Yen et al., 2008). Copepods generally have two primary swimming modes, cruising and jumping (Jiang and Kiørboe, 2011). Cruising is a low-Reynolds-number movement, where the copepod beats its cephalic appendages to both swim and create a feeding current (Tiselius and Jonsson, 1990; van Duren and Videler, 1995; Jiang and Osborn, 2004; Svetlichny et al., 2020). Jumping, meanwhile, involves a sequential kick that quickly propels the copepod, usually to escape predators or capture prey (Jiang and Osborn, 2004; Svetlichny et al., 2020).

Given that viscosity increases as temperature decreases and copepod body size increases as temperature decreases, viscosity may play a role in driving copepod body size alongside temperature. As such, a population of copepods grown in a particular temperature and viscosity should differ in size from their offspring if the offspring are hatched and raised in a different temperature or viscosity. By raising copepod offspring in different conditions from their parents, including cultures of decoupled temperature and viscosity, this study seeks to determine the differential effects of temperature and viscosity on the body size of copepods. It is hypothesized that copepods grown in warm, viscous water will grow larger than copepods grown in warm waters of natural viscosity but smaller than copepods grown in cold water of natural viscosity, as temperature still likely plays a role in determining body size.

## RESULTS

The first copepodites appeared in the 30°C culture 1 day after hatching, while those in the 18°C culture appeared 5 days after hatching (Fig. 1). The first adult copepods appeared in the 30°C culture 10 days after hatching, while adult copepods appeared in the 18°C culture 13 days after hatching. The first gravid female was found in the 30°C culture the same day as the first adult, 13 days

after hatching, while the first gravid female in the 18°C culture was found 14 days after hatching. The 30°C cultures had more adults sampled than the 18°C cultures throughout the experiment.

The copepods grown at 18°C are statistically larger than the 30°C copepods every generation (Fig. 2). Adult copepods in 30°C conditions are on average 13.20% smaller in body length when compared to copepods raised in 18°C conditions, $t(235)=16.154$, $P=6.13\mathrm{E}{-}40$ (Fig. 2A). There is high variance in body size between generations within both temperatures, with some generations being statistically larger or smaller than those before and after (please refer to the Data and resource availability section). Additionally, copepods grown at 30°C are 13.72% thinner than 18°C copepods, $t(235)=14.504$, $P=1.79\mathrm{E}{-}34$ (Fig. 2B). Copepods grown at 30°C are overall smaller in volume than those grown at 18°C, $t(235)=15.398$, $P=1.84\mathrm{E}{-}37$ (Fig. 2C).

The mean body length of the copepods increased between Generations 0 and 1 at 18°C by 10.58%, at 30°C by 9.57%, and at 30°C 18PVP (30°C lineage) by 10.26%. The only condition that did not experience an increase in body length between these generations was 30°C 18PVP (18°C lineage), whose copepods decreased in average body length by 2.24% (Fig. 2A). The body lengths for both experimental 30°C 18PVP cultures were not statistically different in size from Generations 1 to 5 and were therefore combined for future statistical analysis, one-way ANOVA, $F_{(n,\mathrm{d.f.})}=0.03489$, $P=0.8520$.

Copepods raised in 18°C water were statistically longer than 30°C 18PVP copepods, $F_{(n,\mathrm{d.f.})}=421.22$, $P<0.000001$ (Fig. 3A). Cultures at both 30°C and 30°C 18PVP did not significantly differ in body length from each other, $F_{(n,\mathrm{d.f.})}=0.2191$, $P=0.6400$, or across the generations. The exception of this is Generation 4, where the 30°C females were significantly smaller than both PVP-altered viscosity cultures ($P=0.04634$ and $P=0.00030$).

Between Generations 0 and 1, all copepods decreased in prosome width, with 30°C 18PVP (18°C lineage) experiencing the greatest (16.41%) decrease in size (Fig. 2B). As with body length, the

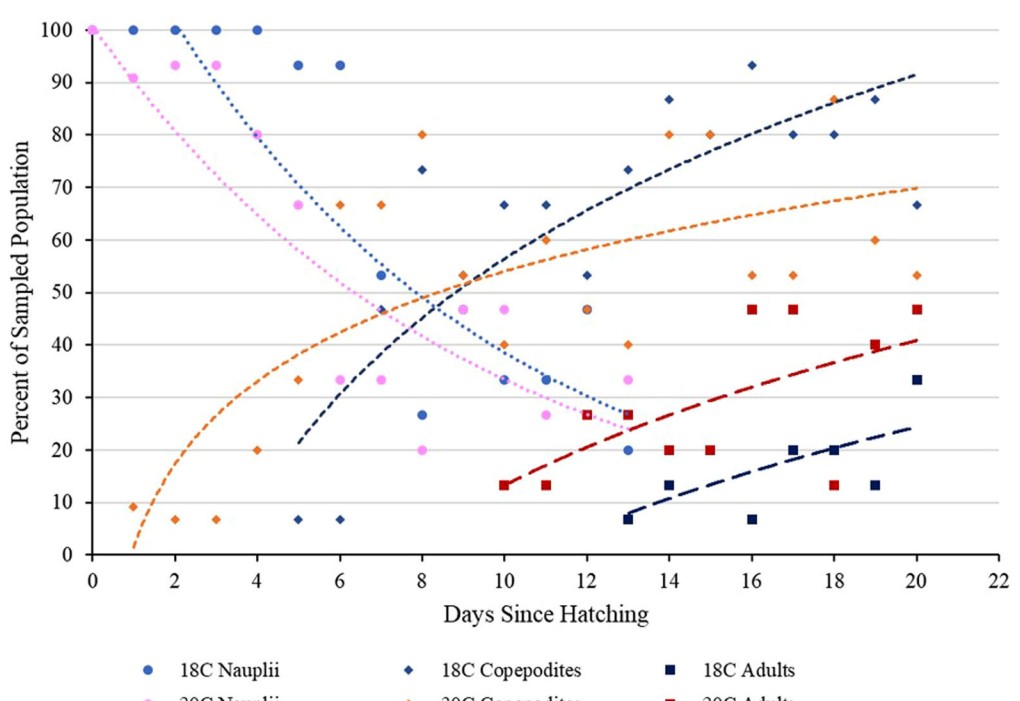

**Fig. 1. Scatterplot of the percent population of nauplii, copepodites, and adults sampled over time from the two stock temperature conditions, 18°C and 30°C (n=15), for each day sampled.** Exponential regression was calculated for the nauplii (18°C $R^2$=0.684, 30°C $R^2$=0.758) while a logarithmic regression was calculated for the copepodites (18°C $R^2$=0.700, 30°C $R^2$=0.561) and the adult copepods (18°C $R^2$=0.419, 30°C $R^2$=0.433).

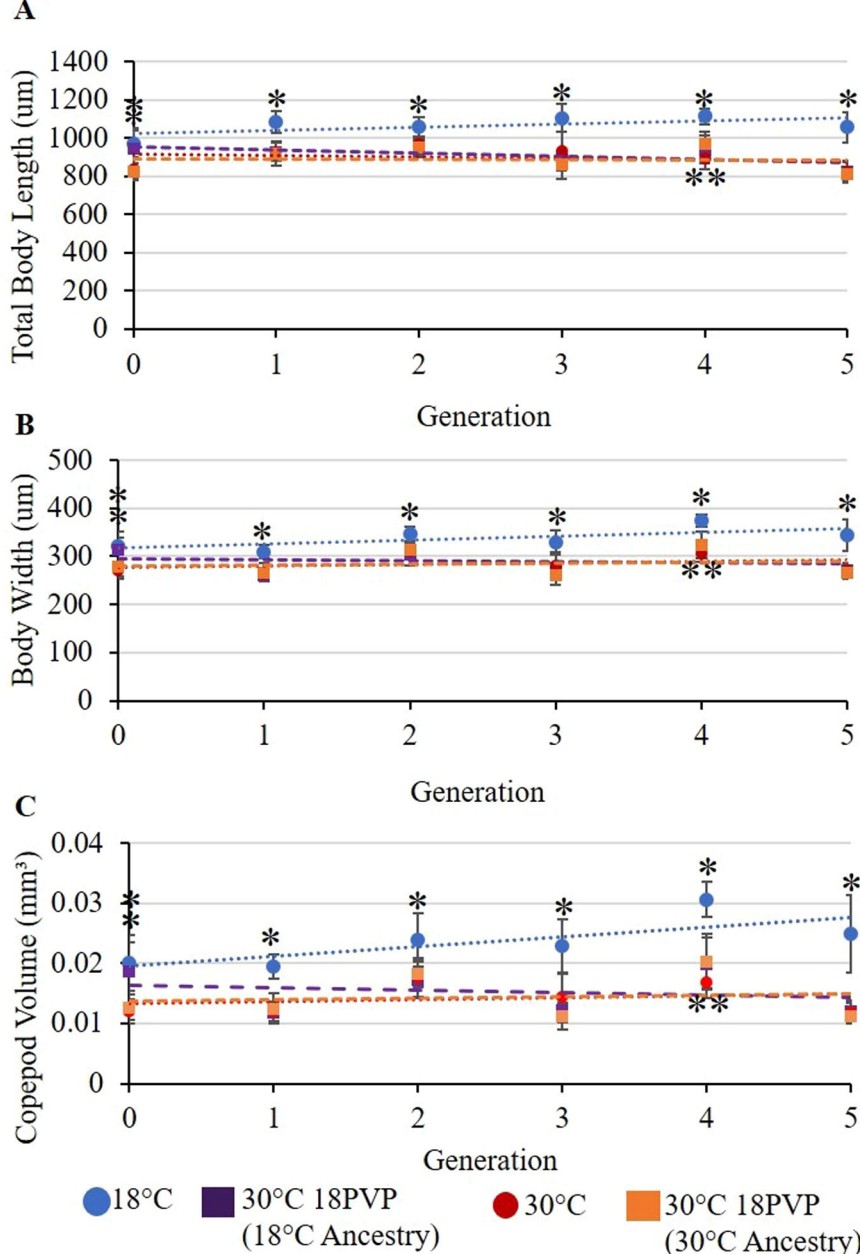

**Fig. 2. Scatter plots with linear regressions of the mean body length, body width and volume.**
(A) body lengths [18°C $R^2$=0.3038, 30°C 18PVP (18°C ancestry) $R^2$=0.1991, 30°C $R^2$=0.0377, 30°C 18PVP (30°C ancestry) $R^2$=0.0021], (B) prosome widths [18°C $R^2$=0.4574, 30°C 18PVP (18°C ancestry) $R^2$=0.0089, 30°C $R^2$=0.0804, 30°C 18PVP (30°C ancestry) $R^2$=0.0143], and (C) calculated body volume [18°C $R^2$=0.557, 30°C 18PVP (18°C ancestry) $R^2$=0.0405, 30°C $R^2$=0.0495, 30°C 18PVP (30°C ancestry) $R^2$=0.0156], comparing *Mesocyclops* copepods across all generations and all test conditions [$n$=20 for all conditions, with the exceptions of Generation 0 18°C, 30°C, and 30°C 18PVP (30°C) $n$=18; Generation 2 18°C and 30°C $n$=19; and Generation 4 30°C 18PVP (18°C) $n$=19]. Single asterisks denote the copepods as significantly larger than others from one-way ANOVA testing, while two asterisks denote that the copepods are significantly smaller than all others.

difference in prosome width between the two 30°C 18PVP cultures was not statistically significant, and these cultures were therefore grouped for analysis, one-way ANOVA, $F_{(n,d.f.)}$=0.02449, $P$=0.8758. Copepods raised in 18°C water were wider than those raised in 30°C 18PVP, one-way ANOVA, $F_{(n,d.f.)}$=202.52, $P$<0.000001, and wider than those raised in 30°C, one-way ANOVA, $F_{(n,d.f.)}$=181.39, $P$<0.00001. As with body length, the prosome width of 30°C copepods and 30°C 18PVP copepods are not significantly different, one-way ANOVA, $F_{(n,d.f.)}$=0.03749, $P$=0.8466.

According to Shapiro-Wilks test, none of the conditions are normally distributed [treatment one-way ANOVA, $F_{(n,d.f.)}$=319.116, $P$<0.001; generation one-way ANOVA, $F_{(n,d.f.)}$=28.919, $P$<0.001]. Using Holm-Sidak testing, 18°C has been shown to be significantly longer than both other test conditions, 18°C versus 30°C mean difference=178.943, $t$(38)=10.740, $P$=4.53E−13; 18°C versus 30°C 18PVP mean difference=178.472, $t$(58)=14.905, $P$=1.7E−21 (Figs 2-4). By Generation 5, the copepods in the 18°C cultures

were 22.29% longer than those in 30°C, ANOVA, $F_{(n,d.f.)}$=115.34, $P$<0.000001, and 22.52% longer than the combined 30°C 18PVP copepods, ANOVA, $F_{(n,d.f.)}$=222.18, $P$<0.000001 (Fig. 4). Additionally, in Generation 5, the copepods grown at 18°C were on average 108.78% larger in volume than the 30°C copepods and were 114.71% larger in volume than the 30°C 18PVP copepods (Fig. 3C). Holm-Sidak testing shows that the 30°C copepods and 30°C 18PVP copepods were not significantly different from one another, 30°C 18PVP versus 30°C mean difference=0.472, $t$(58)=0.192, $P$=0.848. The 30°C 18PVP copepods were 2.99% larger than the 30°C copepods. The 18°C copepods also had a slight positive trend in body size, growing by 0.00019 mm³ with every generation, while both 30°C cultures shrank at less than 0.0001 mm³ every generation (Fig. 2C). The growth measured in the 18°C cultures resulted in copepods that were significantly longer at Generation 5 than those at Generation 0, one-way ANOVA, $F_{(n,d.f.)}$=16.776, $P$=0.000142 (Fig. 3).

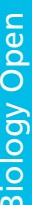

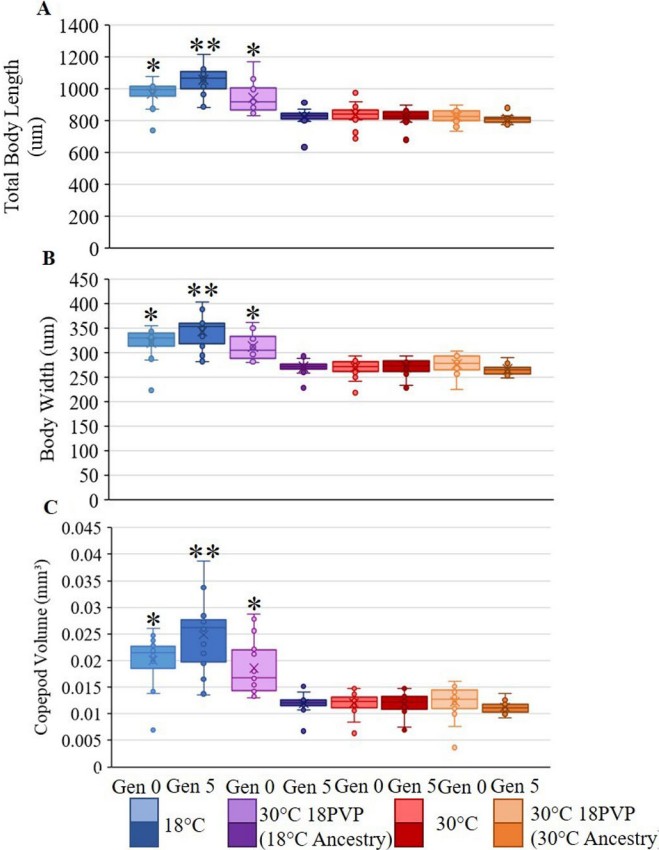

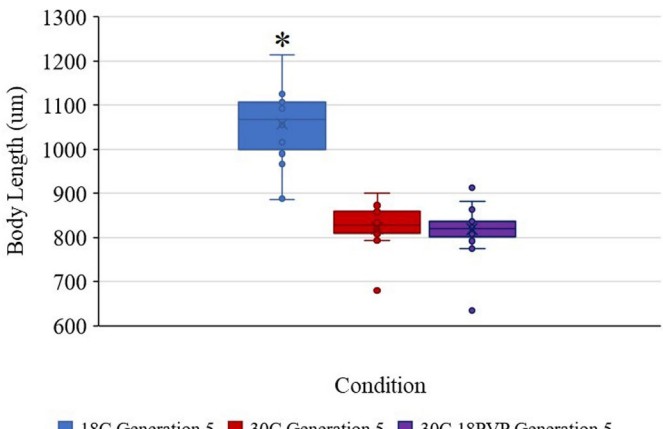

Fig. 4. Box-and-whisker plot comparing the body length of copepods collected from only Generation 5 cultures, including the combined 30°C 18PVP copepods, to highlight the final measured differences in body size (18°C and 30°C *n*=20, 30°C 18PVP *n*=40). The asterisk denotes significance from one-way ANOVA testing.

Fig. 3. Box-and-whisker plot of the (A) body lengths, (B) prosome widths, and (C) calculated body volume comparing *Mesocyclops* copepods at Generation 0 and Generation 5 across all test conditions [*n*=20 for all conditions, with the exceptions of Generation 0 at 18°C, 30°C, and 30°C 18PVP (30°C) *n*=18]. Asterisks denote significant statistical differences determined by one-way ANOVA testing, with Generation 5 18°C copepods being significantly larger than all other copepods and Generation 0 18°C and 30°C 18PVP (18°C ancestry) copepods being smaller than the Generation 5 18°C copepods but larger than all other copepods.

## DISCUSSION

Body size is critical to the ecological function of organisms and to the structure of the food web. Aquatic food webs illustrate a positive correlation between body size and trophic level, with larger consumers typically occupying higher trophic levels and taking advantage of a greater variety of prey than smaller consumers or consumers at lower trophic levels (Petchey et al., 2008; Arim et al., 2010; Potapov et al., 2019). For individual copepods, body size determines not only the size of possible prey but also the maturation rate, number of offspring produced, and the success of offspring (Hopcroft and Roff, 1998; Ramakrishna Rao and Kumar, 2002; Hart and Bychek, 2011). Body size also has an impact on the number of parasites a copepod can host (Wedekind et al., 2000; Van der Veen, 2003). This is critical, as *Mesocyclops* spp. are intermediate hosts for *Dracunculus medinensis*, which causes Guinea worm disease (Bapna, 1985; Bimi, 2007). We found that for *Mesocyclops* spp., temperature is the main driving force behind copepod body size, not viscosity. Copepod body size is highly correlated with the temperature of the water around them, where warmer waters had copepods grow to smaller body sizes, with ancestry, lineage, or viscosity having no effect on the body size of the *Mesocyclops* copepods. This decrease in size in response to a

shift from cold to warm populations additionally occurred within the first experimental generation.

The ratio of total body length to width stayed the same throughout each test condition and generation, indicating that there is dynamic similarity in the noted changes to body size. It is unknown why the copepods on average increased in length between Generations 0 and 1 across all conditions, which resulted in copepods in Generation 1 that were generally longer and thinner than their mothers. The increase in length may be due to the copepods being added to clean jars for the experiment, eliminating potential population pressure and providing clean water, which has been seen in other organisms in aquaculture systems; however, it does not account for the width not matching the changes in length (Suantika et al., 2018; Seo and Park, 2023). Competition is especially problematic with larval crustaceans in high-density systems (Nga et al., 2005; Arnold et al., 2009; Romano and Zeng, 2017). However, the effects of crowding seem to be less prevalent in late-stage crustaceans, such as shrimp in high-density aquaculture systems (Rodríguez-Olague et al., 2021). The interplay of these variables may account for the accumulated growth measured in the 18°C cultures, as both cold temperatures and abundance of food are known to promote copepod growth, but the specifics of these interactions should be researched further (Hart and Bychek, 2011).

While all copepods in 30°C and altered-viscosity conditions were generally of the same size, this trend was broken in Generation 4, where 30°C copepods were significantly smaller in length than copepods in both viscous conditions and were not significantly thinner. Variation between generations is noted within cultures, with populations 'wobbling' from larger to smaller between each generation of organisms (Souissi et al., 2016). It is possible that the size wobble was enough to bring the Generation 4 30°C copepods out of range of the other warm cultures. If the experiment were to continue for further generations, it is likely that this wobble would still be observed, but significant deviations would remain uncommon (Souissi et al., 2016).

While the calculated volumes followed the same trends as the measured lengths and widths, the model used for volume calculation could be made more accurate. The elliptical cone portion overestimates the width of the urosome and, in doing so, overestimates the volume of the copepod. A more sophisticated model would involve more

measurements, such as the depth of the cephalosome and metasome and multiple widths of the prosome and urosome, so that the copepod could be broken into more accurate blocks and shapes. A truly accurate model would involve measurements of each body segment and could further include leg lengths and widths. Measuring the depth of the copepod was not possible in this experiment, as when they were adjusted to their side, the copepods would kick rapidly to reorient themselves. Prior observation has shown that when their muscles relaxed, such as under the effects of MS-222, the prosome would extend in a way not observed in the living *Mesocyclops* sp., and so the measurements would not be applicable to non-anesthetized copepods.

It is well known that increases in temperature decrease an invertebrate's body size or length, which we observed as well (Uye, 1982; Dugan et al., 1994; Garzke et al., 2015; Corona et al., 2021). This is often linked to reproductive timing, as organisms that mature more quickly are often smaller than members of the same species that took longer to mature, with growth slowing down or stopping altogether once sexual maturity is reached (Stamps, 1993; Stamps and Krishnan, 1997). The observed change in maturation time in the Chadian *Mesocyclops* sp. was 10 days after hatching at 30°C and 13 days at 18°C (Ban, 1994; Régnière et al., 2012). Additionally, Chadian *Mesocyclops* sp. females grown in 30°C were smaller than those grown in 18°C, with little overlap between the largest 30°C copepods and the smallest 18°C copepods. Both trends are especially important to invertebrates and other egg-laying organisms, where the size of the female has a direct correlation to her fecundity and, therefore, fitness (Bilgin and Samsun, 2006; Berger et al., 2008). As Chadian *Mesocyclops* have a rapid generation time and shallow ponds and lakes tend to have fast fluxes in temperature, the conditions a parent developed in are likely different than those their offspring will develop in. As such, these changes in body size and reproductive rate are likely an adaptive response to seasonal temperature change (Sasaki et al., 2019; Sasaki and Dam, 2020; de Juan et al., 2023).

While this study concludes that it is temperature that drives copepod body length, not viscosity, further questions arise from this finding. Viscosity may still impact a copepod's ability to cruise through the water, produce feeding flows, and perceive mechanical disturbances in its environment. Future studies may focus on the physical interactions between copepods and high-viscosity environments, as well as comparisons in swimming between the smallest copepod species and the largest copepod species.

## MATERIALS AND METHODS
### Animal selection
*Mesocyclops* from Chad, Africa, were selected to test this relationship between temperature, viscosity, and copepod body size. Lake Chad, and the other bodies of water within the country, can vary between 18.85°C and 36.85°C and, as such, provide a wide range of temperatures to test (Policelli et al., 2018). *Mesocyclops* is a genus of small copepods found nearly globally across the tropics and subtropics, and can handle a wide range of temperatures (Gutiérrez-Aguirre and Suárez-Morales, 2001; Melão and Rocha, 2004; Hołyńska, 2006; Dumont, 2009). *Mesocyclops* have relatively short developmental times. Females reach sexual maturity in less than 3 weeks and produce clutches of eggs as often as every 2 days, allowing for many generations of copepods in a short time period (Gophen, 1976; Hansen and Santer, 1995; Melão and Rocha, 2004; Fereidouni et al., 2015).

### Animal collection and maintenance
Copepods were collected from Chad, Africa, in October 2021 by the Programme National d'Eradication du Ver de Guinée – Tchad, from ponds between Mandélia and Guelendeng, using a thrown hand net. Harvested copepods were sorted into bottles of clean freshwater and shipped to the Georgia Institute of Technology (Atlanta, GIT). The copepods were identified as belonging to the genus *Mesocyclops*, but not to the species level. The *Mesocyclops* spp. were raised in 1 l jars in warmwater baths (25°C) and fed egg yolk diluted 1:5 with water, *ad libitum*, every other day. Cultures were filtered weekly using a 153 μm mesh net to separate adult copepods from nauplii to minimize cannibalism. Adults and nauplii were transferred to clean 1 l jars. Cultures were maintained for 1 year (~26 generations) before adults from the 25°C baths were split into two cultures: one culture raised at 18°C and one at 30°C. The 18°C temperature was maintained by a cold-water bath, chilled by a copper coil in the middle of the bath, with a variation of ±1°C. The 30°C temperature was maintained by a climate-controlled incubator room, set to adjust the temperature when it varied by more than ±1°C. These two new cultures were fed and filtered on the same schedule as the original cultures and maintained for 18 months before experimentation.

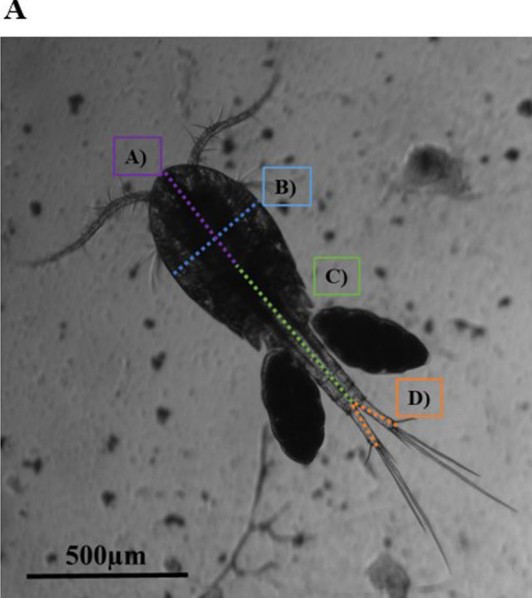
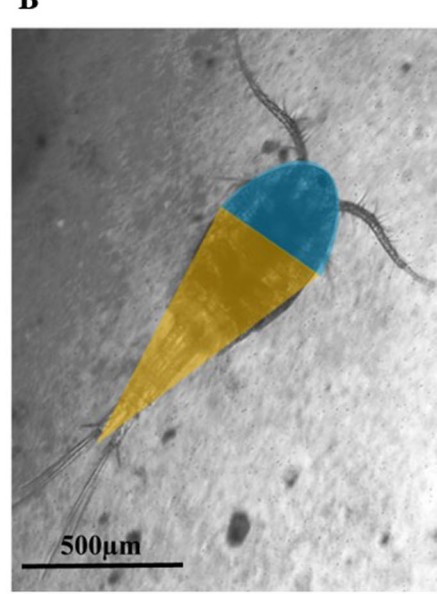

**Fig. 5. Body measurements and visualized model of the *Mesocyclops* spp. copepods.** (A) Measurements of (A) copepod body length down the prosome, (B) prosome width at the widest point, (C) urosome length, and (D) length of the caudal rami. The two caudal ramus measurements were averaged together to create one number. These numbers were recorded individually, with prosome, urosome, and rami lengths being used to create a total body length for each copepod (*n*=473). (B) Visualization of the 'dome and cone' body volume calculation, using a blue dome and a yellow cone.

### Growth rate measurements

To determine the time it takes copepods to mature from a newly hatched nauplius to an adult, 20 adult, gravid female *Mesocyclops* spp. were taken from 18°C and 30°C stock cultures and placed in 200 ml of water at the same temperature at which they were grown. Gravid females were identified by the pair of egg sacs attached to the base of the urosome of the copepod. The gravid *Mesocyclops* spp. were given 24 h to hatch their eggs before being removed, and the hatched nauplii were randomly distributed into four 1 l jars. Female *Mesocyclops* sp. produce an average of 19.98±6.28 eggs at a time (*n*=87), and so a mean estimate of 400 eggs was calculated for the starting number of nauplii in total or, if evenly distributed among the four jars, 100 nauplii per 1 l jar. It was noted as well that not all eggs had hatched at this point.

From the newly established cultures, ten nauplii were randomly selected, measured, and removed from the culture. Afterward, 15 individuals were randomly selected daily from the experimental 18°C and 30°C cultures, staged, measured, and removed to the stock cultures. Fifteen individuals were selected, as that was the mean lowest quartile of eggs produced by female *Mesocyclops* sp., while the mean of the middle quartile was 21 (*n*=87), and so we would sample 75% of the estimated total population. Staging categories were broad to include the major life stages of nauplius, copepodite, and adult copepods. Nauplii are denoted as having a rounded body, a singular eye, and three pairs of cephalic appendages. Copepodites superficially resemble adult copepods in body plan, with distinct cephalosome, metasome, and urosome, with fewer body segments and proportionally shorter antennae than the adults. Adult female *Mesocyclops* copepods are characterized by five body segments, visible lipids in the urosome, and – in gravid individuals – large, paired egg clusters attached to the urosome. Adult males are identified by their geniculate antennae, differentiating them from large female copepodites. This staging and measuring process continued for 20 days. Size measurements were averaged and used to calculate daily growth and developmental rates for the population.

### Copepod selection

Only gravid female copepods were measured throughout the generational study. Gravid female copepods were easy to identify due to the pair of egg sacs attached to the urosome. Since female copepods can only be gravid as adults, selecting gravid females ensures they are all at the same life stage and allows for close comparisons between mothers and daughters on a population level. This technique allowed us to compare size attributes within the lineage over multiple generations. Lastly, and importantly, female *Mesocyclops* are generally larger than males, and as such, the sex sampled needs to be controlled or else sexual selection bias would be introduced to the measurements (Dahms and Fernando, 1995; Pilati and Menu-Marque, 2002; Can and Bozkurt, 2019).

### Body measurements

Measurements were done with a Nikon Ti-E inverted microscope using NIS-Elements software. Individual copepods were photographed and measured using the NIS-Elements software. Prosome length was measured from the tip of the cephalosome to the posterior margin of the metasome. Tail length was recorded from the end of the metasome to the terminal segment of the caudal rami. As the rami are branched, two lines were used, and the lengths were averaged. Measurements did not include setal hairs on the end of the caudal rami. Body width was measured at the widest point of the prosome (Fig. 5A). Once measured, the live copepods were placed in a 200 ml Petri dish of their respective test conditions.

### Volume calculations

Historically, copepod volume is modeled as a sphere, which overestimates the actual volume of the animal. To better approximate the volume of the copepod, we used a combined 'dome and cone' model, where we split the body of the copepod into two parts. The dome is the approximate volume of the cephalosome, while the cone is the volume of the rest of the prosome and the urosome (Fig. 5B). Both the remainder of the prosome and the entire length of the urosome were grouped into the volume calculation of the cone.

As most copepods, *Mesocyclops* included, are dorsoventrally flattened, the total volume of the dome and cone were divided in half to account for the ventral flattening and so create an elliptical cross section. This was calculated through the following modified equation:

$$V = \frac{\left(\frac{\frac{4}{3}\pi r^3}{2}\right) + \left(\frac{\pi r^2 (L - r)}{3}\right)}{2}, \tag{1}$$

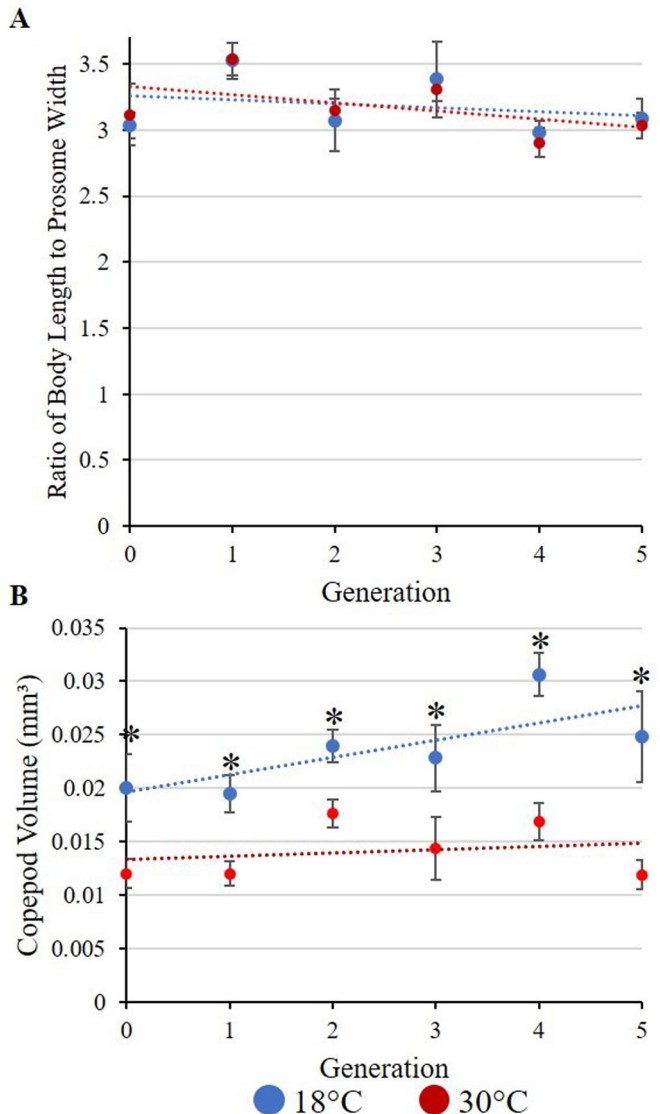

**Fig. 6. Scatterplots of calculated copepod body ratios between length and width, as well as calculated mean body volume.** (A) Scatterplot of the ratio between copepod body length and maximum prosome width between generations (*n*=20 copepods measured per condition per generation, with the exception of 18°C Generation 2, where *n*=18). Higher numbers indicate longer, thinner copepods (18°C $R^2$=0.0655, 30°C $R^2$=0.2663). (B) Scatterplot of mean copepod body size (mm³) across generations and temperature conditions, calculated using the body length and prosome widths fit to cone and dome model (Eqn 1) and fit to a linear trendline (18°C $R^2$=0.557, 30°C $R^2$=0.0495). Significance is denoted by asterisks and is tested within each generation. Note that by the end of the study, the copepods grown at 18°C are on average 2.17 times the volume of the other copepods, but with higher variance. The copepods grown at 18°C also have a slight positive trend in body size, but this is insignificant within the scope of the experiment.

Biology Open

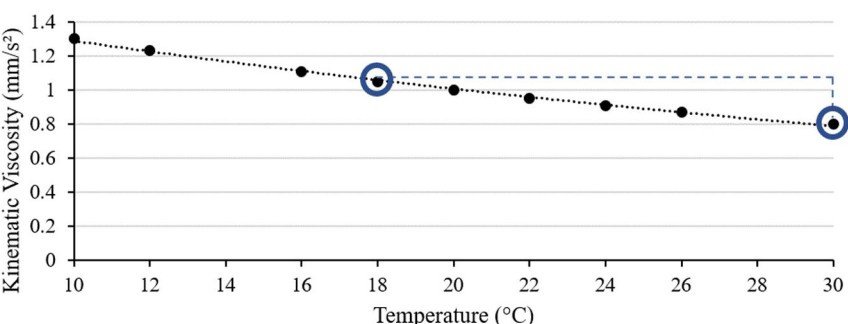

**Fig. 7. Scatterplot of mean freshwater kinematic viscosity measurements (mm/s²) ($n$=3) between 10°C and 30°C, fit to an exponential curve ($R^2$=0.9973).** The two test conditions, 18°C and 30°C, are circled. The 18°C measurements were 1.05 mm/s², while the 30°C measurements were 0.80 mm/s², a 23% decrease in viscosity.

where $V$ is the approximate volume of the copepod, $r$ is half the widest width of the prosome, and $L$ is the total length of the copepod. The cone and dome were summed and subsequently divided by two to create the elliptical volume. The ratio of prosome length to width was monitored over the five generations as a simple measure to track any changes occurring were scaling across several dimensions (Fig. 6A). The volume estimates were then plotted across their generations (Fig. 6B).

**Viscosity measurements**

Viscosity of the water was measured using a Cannon-Fenske 50 glass viscometer according to Cannon's recommended practices. It is important that before measuring, both the viscometer and the fluid are brought to equivalent temperature. Fluid is poured into one end of the viscometer and drawn with a vacuum to a chamber marked on either end. Once the fluid is pulled past the chamber, the vacuum is released, and gravity pulls the fluid

**Fig. 8. Experimental design of the establishment and treatment of *Mesocyclops* spp. cultures across experimental conditions.** Lineages begin at Generation 0 from adults taken from stock cultures and end at Generation 5. Each generation starts with four culture jars of randomly assigned nauplii produced by the previous generation. Hash-marked cultures represent original stocks, blue culture represents 18°C, purple represents 30°C 18PVP, orange represents 30°C 18PVP, and red represents 30°C.

Biology Open

through the chamber. Once the fluid passes the first mark of the chamber, a timer is started and allowed to run until the meniscus of the sample passes the second, lower mark in the viscometer chamber. Common practice has this repeated a minimum of three times per temperature or fluid and averaged. Kinematic viscosity is calculated through the following equation:

$$kv = t \times c \times d, \qquad (2)$$

where $kv$ is the kinematic viscosity in centistokes (mm/s$^2$), $t$ is the time (s) the fluid takes to pass through the chamber, $c$ is the calibration factor of the given viscometer, and $d$ is the density of the fluid (g/cm$^3$).

Viscosity was modified independently of temperature using polyvinylpyrrolidone (PVP; MW 3600 kDa, Sigma-Aldrich 9003-39-8). PVP is a high-molecular-weight neutral sugar that, when mixed with water, increases the viscosity of the water. PVP is also safe for copepods to live in for extended periods of time. The PVP used in this experiment was stored in an oven set to 60°C to keep moisture from adding to the mass measured. Powdered PVP was measured in a plastic weigh boat with an electronic balance in a dry, temperature-controlled room. The massed PVP was added to water to create mixtures of PVP with concentrations between 0.25 and 2.0 g/l, with each mixture measured five times with the viscometer (Fig. 7). Through these measurements, the viscosity of the 30°C cultures was altered from 0.80 mm/s² to match the cooler (18°C) culture's viscosity of 1.05 mm/s², which was found to be 1.98 g/l PVP (Fig. 7). These viscous measurements will be referred to by their associated temperature-viscosity combination, 30°C 18PVP. Three sterilized, opaque, 20 l plastic carboy containers were filled with the 1.98 g/l PVP water mixture and capped to prevent evaporation. These carboys were used as the water reservoir for the 30°C 18PVP cultures. Currently, there are no methods to reduce fluid viscosity independent of temperature without killing copepods. As such, only increases in viscosity independent of temperature were measured in this study.

Gravid females from the 30°C ($n=38$) and 18°C ($n=36$) cultures were measured using the Nikon Ti-E inverted microscope and then placed in labeled Petri dishes with 200 ml of water of their respective test conditions, 18°C ($n=18$), 30°C ($n=20$), and 30°C 18PVP. There were two sets of 30°C 18PVP dishes, one whose parents were from 18°C ($n=18$) cultures, and the other whose parents were from 30°C cultures ($n=18$). Females were given 2 days to hatch their eggs before being removed. The hatched nauplii were stirred and then randomly assorted into their respective experimental group. Each group consisted of four 1 l culture jars kept at the specific temperature and/or viscosity. The cultures were fed every other day, *ad libitum*, with raw egg yolk. The cultures grew for 2 weeks before being filtered using a 150 µm mesh net, with each jar's population within the experimental group being combined into one dish. Between 18 and 20 gravid *Mesocyclops* spp. were removed from each experimental group and measured. The number of measured copepods varied due to the number of mature females at the time of collection. Once the gravid copepods were measured, they were then placed in 200 ml of their test condition water for 2 days. Once their eggs hatched, the adult females were removed, and the cycle began again (Fig. 8). Souissi et al. (2016) examined the effects of temperature on the life history of *Eurytemora affinis* and found that five generations were enough for the copepods to acclimate to their new environmental conditions, with notable changes in the prosome length, egg clutch size, and mortality rate (Souissi et al., 2016). As such, this process was repeated for five generations.

One-way ANOVA tests were used to compare body sizes within the same condition across generations, as well as across conditions within the same generations. Two-way ANOVA tests were done to compare conditions and the effects of generation on copepod anatomy. Key comparisons were made between adjacent generations, the first and last generations, and overall trends. Statistical analysis was performed using Microsoft Excel (v1808) and SigmaPlot (v11.3).

## Acknowledgements
We would like to thank the Programme National d'Eradication du Ver de Guinée – Tchad for their work in collecting the copepods used in this work and for sharing them with the Georgia Institute of Technology for Guinea worm-related research, as well as the Ratcliff Laboratory at the Georgia Institute of Technology for allowing us to use their microscopy system.

## Competing interests
The authors declare no competing or financial interests.

## Author contributions
Conceptualization: Z.W., G.W., D.M.F., J.Y.; Data curation: Z.W., J.Y.; Formal analysis: Z.W., J.Y.; Funding acquisition: J.Y.; Investigation: Z.W., G.W., J.Y.; Methodology: Z.W., G.W., D.M.F., J.Y.; Project administration: D.M.F., J.Y.; Resources: D.M.F., J.Y.; Supervision: D.M.F., J.Y.; Validation: D.M.F., J.Y.; Visualization: Z.W.; Writing – original draft: Z.W.; Writing – review & editing: Z.W., D.M.F., J.Y.

## Funding
We would like to extend special thanks to The Carter Center for funding this research. The Global Campaign to Eradication Dracunculiasis receives financial support from a large coalition of organizations and agencies. Please refer to www. cartercenter.org/ways-to-give/funding-partners/. Open Access funding provided by Georgia Institute of Technology. Deposited in PMC for immediate release.

## Data and resource availability
All relevant data has been deposited in Dryad at the following link: https://doi.org/10.5061/dryad.2bvq83c5c.

## Peer review history
The peer review history is available online at https://journals.biologists.com/bio/lookup/doi/10.1242/bio.062549.reviewer-comments.pdf.

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
