## [Peer Review File · Biology Open]

Separating the Generational Effects of Temperature and Viscosity on the Body Size of a Freshwater Mesocyclops Copepod

Griffin Wagner, David Fields, Jeannette Yen and Zachary Wagner
10.1242/bio.062549

Editor: Valentina di Santo

Review timeline

Original submission:	27 February 2026
Editorial decision:	5 March 2026
First revision received:	30 March 2026
Editorial decision:	2 April 2026
Second revision received:	6 April 2026
Accepted:	7 April 2026

Original submission

First decision letter

MS ID#: bio.062549

MS Title: Separating the Generational Effects of Temperature and Viscosity on the Body Size of a Freshwater Mesocyclops Copepod

Authors: Zachary Wagner; Griffin Wagner; David Fields; Jeannette Yen

I have now reached a decision on the above manuscript.

The reviewer reports are shown at the bottom of this email.

As you will see, the reviewers raised a number of substantial criticisms that prevent me from accepting the paper at this stage.

They suggest, however, that a revised version might prove acceptable, if you can address their concerns. If you think that you can deal satisfactorily with the criticisms on revision, I would be pleased to see a revised manuscript. We would then return it to the reviewers.

At this stage, we also ask you to ensure your manuscript complies with our formatting guidelines. Provided you are able to fully address the referees' comments, we are positive about publication of your paper (we accept over 95% of revision submissions) and therefore hope you won't mind any extra work involved in reformatting your manuscript at this point.

Please upload both a 'clean' version of your Word file, along with a highlighted version clearly showing where you have made changes in the revised manuscript. Please avoid using 'Track changes' in Word files as these are lost in PDF conversion.

I should be grateful if you would also provide a point-by-point response detailing how you have dealt with the points raised by the reviewers in the 'Response to Reviewers' box. Please attend to all of the reviewers' comments. If you do not agree with any of their criticisms or suggestions please explain clearly why this is so.

Reviewer 1

Comments for the author

This manuscript addresses a timely question: how much of body-size variation in *Mesocyclops* is driven by temperature-dependent metabolism versus water viscosity. Following populations across five generations is a real strength, and the underlying dataset is sound. That said, the current framing of the viscosity hypothesis, the assumptions used for volume estimation, and the quality/clarity of the figures need substantial work before I can recommend publication. My specific comments are below.

C1: The authors describe viscosity as a "little studied variable". I recommend a more cautious formulation; the debate regarding thermal versus viscous drivers for plankton size is well-documented, and the authors already cite literature that decouples these effects.

C2: The manuscript frames viscosity as a key driver, but the results don't really support that. The temperature effect on size is clear, whereas the 30°C vs 30°C+PVP comparison appears largely non-significant (aside from an isolated generational difference). As written, the narrative feels over-weighted toward viscosity relative to what the data show. Please either (i) revise the framing and conclusions accordingly, or (ii) provide stronger evidence that viscosity has an independent effect.

C3: I can't reproduce the viscosity manipulation from the current Methods. You mention "known concentrations of PVP" but don't state the concentration (units), how it was prepared, or whether viscosity was verified in the actual culture jars over time. Please add the full recipe and measurement protocol.

C4: The dome+cone approximation (eq. 1) assumes a circular cross-section derived from width. For copepods, dorsoventral flattening can be non-trivial, so relying on width alone may bias volume estimates. If you're going to keep this model, please justify the assumption (or estimate the error). Alternatively, use an elliptical cross-section or include a depth measurement.

C5: I suggest that the authors clarify if the urosome volume was calculated as part of the primary cone or if a multi-segment model—calculating the prosome and urosome volumes independently—would have provided a more precise estimate, especially given the massive volume differences reported.

C6: The "Growth rate measurements" section does not allow for full reproducibility. Since 15 individuals were removed daily, please clarify the total starting population to ensure that density-dependent growth factors were controlled.

C7: While you mention that copepods were "staged", you don't define the specific morphological markers used to distinguish between different life stages. I think that providing the criteria for staging nauplii versus copepodites is essential.

C8: This is a minor comment: The definition of maturity based on the presence of egg sacs (gravid females) is practical but narrow. It may overlook the period between the final moult and first reproduction (isn't that?). Please clarify if "adult" status was confirmed anatomically or solely by egg production.

C9: The manuscript doesn't specify if measurements were performed on live or preserved individuals. If preserved, please comment on potential shrinkage and its effect on the absolute body size measurements reported.

C10: Re the figures: The current presentation of the figures is a bit too insufficient. The use of placeholders within the text and the separation of related panels (particularly in Figs 1,5,6) make the data difficult to navigate.

Figure 1: The line annotations (A-D) overlap with anatomical features. These must be redrawn for clarity, and a scale bar must be added to provide a sense of absolute size.

Figure 5: Panels 5A,B,C are fragmented. These should be merged into a single composite figure with standardised x and y-axes to track the population shift over time accurately.

In Figs 6 and 7, the y-axis ranges are inconsistent. Standardising these ranges is necessary for easier comparison.

C11: Suppl Materials. I couldnt access this file as it was referring me to an external link. I recommend providing the raw measurement data (individual lengths and widths) as a spreadsheet in addition to the provided plots.

C12: The authors affiliations are incomplete.

Reviewer 2

Comments for the author

This is a clear, comprehensive write-up of a well-designed experiment seeking to parse the relative contributions of temperature and viscosity on body size, finding that viscosity alone has little effect. The methods are sufficiently detailed and substantiated, and the conclusions are drawn directly from the results. I have a few recommended corrections, which constitute improvements to figures and minimal clarifications in the text. Below I address each category required in the reviewer rubric guidelines:

Experimental quality

The quality of the experimental design and written methods is high. Appropriate statistical testing is used throughout. Care has been taken to perform experiments in such a way that the hypothesis is being tested without the potential to conflate intersecting factors such as by only sampling gravid females. Experimental decisions are well-reasoned. Every figure is very clear and all statistics are included either in the figures themselves or in the text.

Reproducibility

The methods are appropriately detailed and permit reproducibility. The authors have included a good level of detail of all measurements taken and statistical tests performed.

The number of sampled individuals is appropriate, though I think it would be more rigorous to cite similar studies that use similar sampling numbers, or substantiate the sample number. Why were 10 nauplii selected vs 15 copepods? How were these sample numbers chosen?

Similarly, why were five generations chosen? Would more generations clarify the effect of 'wobbling' between generations? If so this could be mentioned in the discussion (lines 295-300).

Completeness

The conclusions are supported by the data. I do not think there are any flaws in the experimental design that would invalidate any of the results. The authors explain why lower viscosities could not be tested. It would be interesting to try a larger range or number of viscosities but that is by no means necessary within this particular study.

Scholarship

I have no concerns regarding the scholarship, the manuscript is appropriately cited and appropriate studies are discussed.

I have a few additional comments, principally concerning figures.

Lines 271-274 "Body size also has an impact on the number of parasites a copepod can host (Wedekind et al. 2000, Van der Veen 2003). This is critical, as Mesocyclops spp. are intermediate hosts for Dracunculus medinensis, which causes Guinea worm disease (Bapna 1985, Bimi 2007)."

This point is worth expanding on. Critical in what sense? I can infer this but it should be spelled out explicitly in the text. This should be mentioned in the introduction and then commented on in the discussion.

Figure 1

A scale bar is required

Figure 2

Figure 2 is not referred to in the text.

Figure 6

The caption should reiterate the statistical test that was performed to generate the regression lines, similar to how it states that One-Way ANOVAs were performed.

The caption needs to be more explicit about what the asterisks mean, or different colours/symbols need to be used to clarify which tests refer to which asterisks (see comment on figure 7).

This could benefit from different symbols in addition to different colours. I would also use different lengths of dashes for the lines, the red, orange and purple may be difficult to distinguish.

Figure 7

Markers denoting significance through One-Way ANOVA testing could be clearer. From the figure alone I would assume that all three categories on the left are recovered as significantly different to every other category, and that the double asterisk indicates that the the generation 5 copepods are also significantly different to the generation 0 copepods at 18 degrees and 30 degrees 18PVP but this needs to be stated in the caption more explicitly than "asterisks denote significant statistical difference..."

Reviewer's Responses to Questions

Experimental quality

Does each figure have the proper controls?

If 'No', please indicate reasons in Comments for Author box below.

Reviewer #1:

- Yes

Reviewer #2:

- Yes

Were the data analyzed using appropriate statistical tests?

If 'No', please indicate reasons in Comments for Author box below.

Reviewer #1:

- Yes

Reviewer #2:

- Yes

Reproducibility

Were experiments performed using adequate number of biological replicates?
If 'No', please indicate reasons in Comments for Author box below.

Reviewer #1:

- Yes

Reviewer #2:

- Yes

Does the methods section provide sufficient detail to permit reproducibility?
If 'No', please indicate reasons in Comments for Author box below.

Reviewer #1:

- No

Reviewer #2:

- Yes

Completeness

Are the manuscript's conclusions supported by the data?
If 'No', please indicate reasons in Comments for Author box below.

Reviewer #1:

- Yes

Reviewer #2:

- Yes

Scholarship

Do the authors cite and discuss the merits of data that would argue for and against their conclusion?

If 'No', please indicate reasons in Comments for Author box below.

Reviewer #1:

- Yes

Reviewer #2:

- Yes

Does the manuscript title & abstract accurately reflect the contents of the manuscript, without hyperbole?

If 'No', please indicate reasons in Comments for Author box below.

Reviewer #1:

- Yes

Reviewer #2:

- Yes

First revision

Author response to reviewers' comments

Our responses are written in the green text.

Reviewer 1: This manuscript addresses a timely question: how much of body-size variation in *Mesocyclops* is driven by temperature-dependent metabolism versus water viscosity. Following populations across five generations is a real strength, and the underlying dataset is sound. That said, the current framing of the viscosity hypothesis, the assumptions used for volume estimation, and the quality/clarity of the figures need substantial work before I can recommend publication. My specific comments are below.

C1: The authors describe viscosity as a "little studied variable". I recommend a more cautious formulation; the debate regarding thermal versus viscous drivers for plankton size is well-documented, and the authors already cite literature that decouples these effects.

We have replaced "little studied variable" with "key variable to copepod ecology"

C2: The manuscript frames viscosity as a key driver, but the results don't really support that. The temperature effect on size is clear, whereas the 30°C vs 30°C+PVP comparison appears largely non-significant (aside from an isolated generational difference). As written, the narrative feels over-weighted toward viscosity relative to what the data show. Please either (i) revise the framing and conclusions accordingly, or (ii) provide stronger evidence that viscosity has an independent effect.

We thank the reviewer for this observation. While the literature extensively documents the ecological importance of viscosity for small aquatic organisms, previous studies largely focus on behavioral compensations. In contrast, our study investigated the potential morphological impacts—specifically body size—over five generations.

By decoupling the temperature-viscosity link, our experimental design isolated the effects of increased viscosity on copepod size. Given that these organisms operate at the transition between inertial and viscous regimes, the lack of measurable morphological effects over five generations is a significant finding. Our results underscore that temperature remains the primary driver of body size in this species. To address the reviewer's point, we have added a clarifying statement (Line 339) to emphasize that viscosity did not influence morphology in this context. Further evidence supporting the role of temperature in governing growth and size can be found in Lines 379-394.

C3: I can't reproduce the viscosity manipulation from the current Methods. You mention "known concentrations of PVP" but don't state the concentration (units), how it was prepared, or whether viscosity was verified in the actual culture jars over time. Please add the full recipe and measurement protocol.

Thank you for the feedback! We added a paragraph on how to use the glass viscometer, as well as the equation used to calculate kinematic viscosity. We also added a segment on how we created the viscosity conditions, and how we stored the viscous water throughout the course of the experiment.

C4: The dome+cone approximation (eq. 1) assumes a circular cross-section derived from width. For copepods, dorsoventral flattening can be non-trivial, so relying on width alone may bias volume estimates. If you're going to keep this model, please justify the assumption (or estimate the error). Alternatively, use an elliptical cross-section or include a depth measurement.

The dorsoventral flattening is an excellent point, we have updated the equation and the text to account for the elliptical cross-section of the approximation. We have also updated all volume figures.

C5: I suggest that the authors clarify if the urosome volume was calculated as part of the primary cone or if a multi-segment model—calculating the prosome and urosome volumes independently—would have provided a more precise estimate, especially given the massive volume differences reported.

We included the urosome length in the cone height calculations, and specified as such. You bring up a good point, in that this inclusion likely overestimates the overall volume of the copepods, and we added a paragraph about this to the discussion. We postulated a more accurate model that could be implemented in the future, and what measurements one would need to create that model.

C6: The "Growth rate measurements" section does not allow for full reproducibility. Since 15 individuals were removed daily, please clarify the total starting population to ensure that density-dependent growth factors were controlled.

We have added more information for the starting number. The starting number of nauplii was not counted, instead estimated from a previous count of the mean number of eggs each female *Mesocyclops* sp. carried, an average of 19.98 ± 6.32 (n=87), for an estimated starting density of 100 nauplii per jar, and 400 nauplii per condition. We have now included this in the methods section of the paper.

C7: While you mention that copepods were "staged", you don't define the specific morphological markers used to distinguish between different life stages. I think that providing the criteria for staging nauplii versus copepodites is essential.

We have now added a clarifying sentence for each life stage. The copepods were broadly staged into three categories, nauplius, copepodite, and copepod. Nauplii were defined by their round body and six swimming legs. Copepodites were defined by their distinct body segments and small size. Adult males were identified by the number of body segments and their geniculate antennae, and only identified to distinguish them from large female copepodites. Adult female *Mesocyclops* were the target of the study, and so they were the only adult copepods measured. They were identified with their visible lipid bubbles in the urosome, as well as clusters of eggs attached to the urosome. Adult males were identified in the growth portion of the study through their more geniculate antennae.

C8: This is a minor comment: The definition of maturity based on the presence of egg sacs (gravid females) is practical but narrow. It may overlook the period between the final moult and first reproduction (isn't that?). Please clarify if "adult" status was confirmed anatomically or solely by egg production.

Addressed by the above edits, but female copepods have visible lipid bubbles in their urosome, and when they are gravid, they have a pair of egg clusters on either side of the urosome.

C9: The manuscript doesn't specify if measurements were performed on live or preserved individuals. If preserved, please comment on potential shrinkage and its effect on the absolute body size measurements reported.

We added a clarifying sentence to confirm that the copepods were measured live, and that measurements occurred prior to being used to start the next experimental generation.

C10: Re the figures: The current presentation of the figures is a bit too insufficient. The use of placeholders within the text and the separation of related panels (particularly in Figs 1,5,6) make the data difficult to navigate.

We have moved the figures to their appropriate place within the main text. We left the figure citations below, according to BiO's formatting guidelines.

Figure 1: The line annotations (A-D) overlap with anatomical features. These must be redrawn for clarity, and a scale bar must be added to provide a sense of absolute size.

We redrew the lines so that they are not solid, and also made the annotation boxes clear. We have rearranged the annotation boxes so that they are not obscuring any anatomical features. Additionally, we added a 500µm scale bar in the bottom-left corner of both images.

Figure 5: Panels 5A,B,C are fragmented. These should be merged into a single composite figure with standardised x and y-axes to track the population shift over time accurately.

We have now merged the three figures into one figure, and updated the references in the text to refer to the single, combined figure. We also updated the x-axis to include more dates. Additionally, we used trendlines with different dashes to more clearly denote between nauplius, copepodite, and adult trends.

In Figs 6 and 7, the y-axis ranges are inconsistent. Standardising these ranges is necessary for easier comparison.

Thank you for the feedback! We have adjusted them to start the y axis at zero. However, if we extended the y-axis of the width graph to match that of the length graph, it would be difficult to see the differences in body width.

C11: Suppl Materials. I couldnt access this file as it was referring me to an external link. I recommend providing the raw measurement data (individual lengths and widths) as a spreadsheet in addition to the provided plots.

Thank you for letting us know, we have compiled the measurement spreadsheets into one Excel document, and will upload it along with the other files for the paper. It includes the lengths, widths, volume calculations, and the ANOVA for the graphs.

C12: The authors affiliations are incomplete.

We have updated the affiliations to include the corresponding author's email address (Zachary Wagner). If there is more specific information needed for the authors, please let us know.

Reviewer 2: This is a clear, comprehensive write-up of a well-designed experiment seeking to parse the relative contributions of temperature and viscosity on body size, finding that viscosity alone has little effect. The methods are sufficiently detailed and substantiated, and the conclusions are drawn directly from the results. I have a few recommended corrections, which constitute improvements to figures and minimal clarifications in the text. Below I address each category required in the reviewer rubric guidelines:

Experimental quality

The quality of the experimental design and written methods is high. Appropriate statistical testing is used throughout. Care has been taken to perform experiments in such a way that the hypothesis is being tested without the potential to conflate intersecting factors such as by only sampling gravid females. Experimental decisions are well-reasoned. Every figure is very clear and all statistics are included either in the figures themselves or in the text.

Reproducibility

The methods are appropriately detailed and permit reproducibility. The authors have included a good level of detail of all measurements taken and statistical tests performed.

The number of sampled individuals is appropriate, though I think it would be more rigorous to cite similar studies that use similar sampling numbers, or substantiate the sample number. Why were 10 nauplii selected vs 15 copepods? How were these sample numbers chosen?

We chose 15 individuals daily during the growth experiments as that is the mean within lowest quartile of egg production, with the overall mean being 20 eggs per female, and so we would sample about 75% of the mean estimated total population. We have added a segment about this in the methods section.

Similarly, why were five generations chosen? Would more generations clarify the effect of 'wobbling' between generations? If so this could be mentioned in the discussion (lines 295-300).

Five generations were chosen due to a paper by Soussi et al. (2016) in which they did a multigenerational temperature experiment with the copepods *Eurytemora affinis*, and found that after 5 generations, their copepods had reacted to the heat stress of a new environment through different adjustments in prosome length, egg clutch size, fecundity rates, lipid storage, and mortality rates. As for the wobble, we hypothesize that variations in the mean body size would continue throughout the experiment no matter the amount of generations. We added both of these explanations to the methods and the discussion sections.

Completeness

The conclusions are supported by the data. I do not think there are any flaws in the experimental design that would invalidate any of the results. The authors explain why lower viscosities could not be tested. It would be interesting to try a larger range or number of viscosities but that is by no means necessary within this particular study.

Scholarship

I have no concerns regarding the scholarship, the manuscript is appropriately cited and appropriate studies are discussed.

I have a few additional comments, principally concerning figures.

Lines 271-274 "Body size also has an impact on the number of parasites a copepod can host (Wedekind et al. 2000, Van der Veen 2003). This is critical, as *Mesocyclops* spp. are intermediate hosts for *Dracunculus medinensis*, which causes Guinea worm disease (Bapna 1985, Bimi 2007)."

This point is worth expanding on. Critical in what sense? I can infer this but it should be spelled out explicitly in the text. This should be mentioned in the introduction and then commented on in the discussion.

Good suggestion, we should have been more explicit. We moved the first mention of Guinea worm disease into the Methods section, when picking *Mesocyclops* as a target organism for this study, and stated that *Mesocyclops* body size may impact the amount of larvae it can transmit to humans. In the Discussion, we reiterate the relationship between *Mesocyclops* and *Dracunculus medinensis*, and emphasize that larger copepods may be eating more larvae, and therefore being exposed to more larvae and are able to transmit more larvae to humans.

Figure 1

A scale bar is required

We added a scale bar, and we additionally made the measurement lines dotted on the body, so that they obscure less of the image.

Figure 2

Figure 2 is not referred to in the text.

Thank you for catching that! We now have included it in the volume section of the methods.

Figure 6

The caption should reiterate the statistical test that was performed to generate the regression lines, similar to how it states that One-Way ANOVAs were performed.

The caption needs to be more explicit about what the asterisks mean, or different colours/symbols need to be used to clarify which tests refer to which asterisks (see comment on figure 7).

This could benefit from different symbols in addition to different colours. I would also use different lengths of dashes for the lines, the red, orange and purple may be difficult to distinguish.

The suggestion for different symbols is a very good idea, we changed the symbols so that the normal environmental conditions are round, and the altered viscosity conditions are square. Additionally, we kept the trendline for the normal environmental conditions as a small dotted line, and used dashes for the altered viscosity, conditions. We also updated the caption to better explain the differentiation, with larger copepods denoted by a single asterisk, while two asterisks denote smaller copepods.

Figure 7

Markers denoting significance through One-Way ANOVA testing could be clearer. From the figure alone I would assume that all three categories on the left are recovered as significantly different to every other category, and that the double asterisk indicates that the the generation 5 copepods are also significantly different to the generation 0 copepods at 18 degrees and 30 degrees 18PVP but this needs to be stated in the caption more explicitly than "asterisks denote significant statistical difference..."

Thank you for the suggestion, we increased the size of the asterisks for the significantly different copepod groups, and also updated the caption to differentiate between the largest copepods, Generation 5 18°C copepods, and the next largest copepods, Generation 0 18°C and Generation 0 30°C 18PVP (18°C ancestry).

Second decision letter

MS ID#: bio.062549

MS Title: Separating the Generational Effects of Temperature and Viscosity on the Body Size of a Freshwater Mesocyclops Copepod

Authors: Zachary Wagner; Griffin Wagner; David Fields; Jeannette Yen

I have now reached a decision on the above manuscript.

The reviewer reports are shown at the bottom of this email.

As you will see, the reviewers gave favourable reports, but raised some minor points that will however, require amendments to your manuscript before I formally accept it. I hope that you will be able to carry these out, because we would like to be able to accept your paper soon.

At this stage, we also ask you to ensure your manuscript complies with our formatting guidelines - please see our manuscript preparation guidelines for details. Provided you are able to fully address the referees' comments, we are positive about publication of your paper (we accept over 95% of revision submissions) and therefore hope you won't mind any extra work involved in reformatting your manuscript at this point.

Please upload both a 'clean' version of your Word file, along with a highlighted version clearly showing where you have made changes in the revised manuscript. Please avoid using 'Track changes' in Word files as these are lost in PDF conversion.

I should be grateful if you would also provide a point-by-point response detailing how you have dealt with the points raised by the reviewers in the 'Response to Reviewers' box. Please attend to all of the reviewers' comments. If you do not agree with any of their criticisms or suggestions please explain clearly why this is so.

Reviewer 1

Comments for the author

The reviewer appreciates the authors' effort in addressing the comments. This is a solid revision that addresses most of my previous concerns. The manuscript is now much clearer, and the additions to the Methods (specifically the viscosity manipulation and staging criteria) significantly improve reproducibility.

That said, there are still a few issues that needs the authors' attention:

C1: There are a few formatting issues in the manuscript that need to be cleaned up. For example, there is still a visible internal comment in the volume section.

C2: There is a problem with reporting statistical results. For example, the t-test is reported as "t- = 5.69E-10". There are quite a few cases of this throughout the text. I am not sure what t- is here, but the standard way of reporting t-test is: t(df) = value, p = value

C3: The volume equation (Eq. 1) needs correct and clarification. I understand that the final division by 2 may be intended to account for the copepod being effectively flat on the ventral side (i.e. modelling only the dorsal half of the body - am I right?), but this should be stated explicitly in the text. I don't see that this has been done.

More importantly, the dome term appears to be incorrect as written. The expression uses $((w/2)^2)$, whereas a volume term should scale with the cube of a length (i.e. (r^3)).

I recommend revising Eq. 1 to ensure the geometry is correctly defined and clearly explained. Perhaps the authors also need ot check the values?

C4: "discernable" should be "discernible".

Reviewer's Responses to Questions

Experimental quality

Does each figure have the proper controls?

If 'No', please indicate reasons in Comments for Author box below.

Reviewer #1:

- Yes

Were the data analyzed using appropriate statistical tests?

If 'No', please indicate reasons in Comments for Author box below.

Reviewer #1:

- Yes

Reproducibility

Were experiments performed using adequate number of biological replicates?

If 'No', please indicate reasons in Comments for Author box below.

Reviewer #1:

- Yes

Does the methods section provide sufficient detail to permit reproducibility?

If 'No', please indicate reasons in Comments for Author box below.

Reviewer #1:

- Yes

Completeness

Are the manuscript's conclusions supported by the data?

If 'No', please indicate reasons in Comments for Author box below.

Reviewer #1:

- Yes

Scholarship

Do the authors cite and discuss the merits of data that would argue for and against their conclusion?

If 'No', please indicate reasons in Comments for Author box below.

Reviewer #1:

- Yes

Does the manuscript title & abstract accurately reflect the contents of the manuscript, without hyperbole?

If 'No', please indicate reasons in Comments for Author box below.

Reviewer #1:

- Yes

Second revision

Author response to reviewers' comments

Reviewer 1: The reviewer appreciates the authors' effort in addressing the comments. This is a solid revision that addresses most of my previous concerns. The manuscript is now much clearer, and the additions to the Methods (specifically the viscosity manipulation and staging criteria) significantly improve reproducibility.

That said, there are still a few issues that needs the authors' attention:

C1: There are a few formatting issues in the manuscript that need to be cleaned up. For example, there is still a visible internal comment in the volume section.

Thank you for letting us know, we have deleted the comment from the manuscript, and also removed the other internal edits remaining.

C2: There is a problem with reporting statistical results. For example, the t-test is reported as "t- = 5.69E-10". There are quite a few cases of this throughout the text. I am not sure what t- is here, but the standard way of reporting t-test is: t(df) = value, p = value

Thank you very much for the feedback, we have made the corrections so that they are in their standard forms.

C3: The volume equation (Eq. 1) needs correct and clarification. I understand that the final division by 2 may be intended to account for the copepod being effectively flat on the ventral side (i.e. modelling only the dorsal half of the body - am I right?), but this should be stated explicitly in the text. I don't see that this has been done.

More importantly, the dome term appears to be incorrect as written. The expression uses $((w/2)^2)$, whereas a volume term should scale with the cube of a length (i.e. (r^3)).

I recommend revising Eq. 1 to ensure the geometry is correctly defined and clearly explained. Perhaps the authors also need ot check the values?

Thank you for the suggestions! We accidentally squared instead of cubed the dome, that was a good catch. We corrected that in the formula, and have updated the values and remade the figures to reflect the accurate values. This didn't change any of the trends or significance, but it did slightly change the values. We also updated the captions where appropriate. We re-wrote the

equation to the standardized geometric units, and defined “ r ” in the text as “half the widest width of the prosome” and removed “ w ” from the equation.

C4: "discernable" should be "discernible".

Good catch, we made the spelling correction!

Third decision letter

MS ID#: bio.062549

MS Title: Separating the Generational Effects of Temperature and Viscosity on the Body Size of a Freshwater Mesocyclops Copepod

Authors: Zachary Wagner; Griffin Wagner; David Fields; Jeannette Yen

I am happy to tell you that your manuscript has been accepted for publication in Biology Open, pending our standard publication integrity checks. It was accepted on 7th April 2026.